# Decorated bacteria-cellulose ultrasonic metasurface

Zong-Lin Li[1,2,4], Kun Chen [3,4], Fei Li [2,4], Zhi-Jun Shi[3], Qi-Li Sun[1], Peng-Qi Li[2], Yu-Gui Peng [1], Lai-Xin Huang[2], Guang Yang [3] ✉, Hairong Zheng [2] ✉ & Xue-Feng Zhu [1] ✉

Cellulose, as a component of green plants, becomes attractive for fabricating biocompatible flexible functional devices but is plagued by hydrophilic properties, which make it easily break down in water by poor mechanical stability. Here we report a class of $SiO_2$-nanoparticle-decorated bacteria-cellulose meta-skin with superior stability in water, excellent machining property, ultrathin thickness, and active bacteria-repairing capacity. We further develop functional ultrasonic metasurfaces based on meta-skin paper-cutting that can generate intricate patterns of ~10 μm precision. Benefited from the perfect ultrasound insulation of surface Cassie-Baxter states, we utilize meta-skin paper-cutting to design and fabricate ultrathin (~20 μm) and super-light (<20 mg) chip-scale devices, such as nonlocal holographic meta-lens and the 3D imaging meta-lens, realizing complicated acoustic holograms and high-resolution 3D ultrasound imaging in far fields. The decorated bacteria-cellulose ultrasonic metasurface opens the way for exploiting flexible and biologically degradable metamaterial devices with functionality customization and key applications in advanced biomedical engineering technologies.

Cellulose is a hydrophilic structural material that vastly exists in the cell walls of green plants and shows excellent mechanical property[1,2], good biodegradability, and high porosity[3], making it the best candidate for paper and other fabrics[4]. For centuries, inspired by the paper-cutting art, the techniques of paper-cutting and folding were used to fashion cellulose thin films (or sheets) into complex structures with unconventional mechanical and morphological responses[5–7], further leading to the implementation of biocompatible flexible devices[8,9]. However, owing to the strong hydrophilic nature and water-induced swelling[10], the cellulose film in a water environment suffers much from poor mechanical stability and structural collapses[11], restricting its potential applications in underwater flexible electronics as well as soft robots[12–16]. Modifying materials with superhydrophobic interfaces is a feasible approach to avoid the influence of the wetting surface[17]. It has been kept on exploring various interesting superhydrophobic

interfaces which support anti-wetting property. In previous works, researchers have developed physical and chemical methods to prepare superhydrophobic surfaces, such as laser and template etching, electrochemical deposition and chemical etching, etc.[18,19]. Those methods can be classified into two categories: increasing the surface roughness and lowering the surface energy via chemical modification. However, the preparation of superhydrophobic cellulose film has faced the following problems. The first is that the physical etching can hardly be employed on the delicate cellulose film. The second is that cellulose fibers are hydrophilic in nature instead of being hydrophobic. Also, it is very challenging to realize large-area, durable and low-cost superhydrophobic surfaces with good mechanical processability[18,19]. To overcome these challenges, we resort to revolutionizing the current paradigm. First, robust superhydrophobic shielding must be introduced by decorating the cellulose fibers with nanoparticles to largely

[1]School of Physics and Innovation Institute, Huazhong University of Science and Technology, 430074 Wuhan, China. [2]Shenzhen Institute of Advanced Technology, and Biomedical Imaging Science and System Key Laboratory, Chinese Academy of Sciences, 518055 Shenzhen, China. [3]College of Life Science and Technology, Huazhong University of Science and Technology, 430074 Wuhan, China. [4]These authors contributed equally: Zong-Lin Li, Kun Chen, Fei Li. ✉e-mail: yang_sunny@yahoo.com; hr.zheng@siat.ac.cn; xfzhu@hust.edu.cn

increase the surface roughness and meanwhile reduce the surface energy. Second, the fabricated superhydrophobic cellulose film should be compatible with high-precision laser cutting for hollow-out acoustic metasurfaces of functionality customization.

Here we present a unique class of $SiO_2$-nanoparticle-decorated bacteria-cellulose (BC) meta-skins to achieve cellulose-based paper-cutting functional metamaterials in water environment. The BC meta-skin has a 3D fibrous network bonded with $SiO_2$ nanoparticles, which renders excellent superhydrophobicity and robust stability (Supplementary Movies 1 and 2). For example, the contact angle (CA) of a water droplet on the metasurface immersed in water for over 200 days is still over 150° (Supplementary Fig. 1), and the superhydrophobic effect is still maintained after self-healing (Supplementary Movie 3). In addition, the BC meta-skin is ultrathin and ultralight, with the thickness being only 20 μm and one piece of a chip-scale film ($1 \times 1$ cm$^2$) weighing only 7.6 mg (Supplementary Movie 4). The fibrous structure has an intriguing bacteria-repairing capacity (Supplementary Fig. 2), making the BC meta-skin a unique candidate for bio-active metamaterials. We find out that the BC meta-skin is well suited for paper-cutting, which can be finely processed by the laser-cutting technology to generate complicated hollow-out patterns, enabling the precision of ~10 μm. Previous efforts of metamaterial paper-cutting mainly focused on creating materials and/or structures with unique mechanical and/or morphological responses[20,21]. However, due to the existence of stable Cassie-Baxter states (viz., micro-sized air cavities) on the nanosurface of BC meta-skins, we hereby focus on the acoustic properties of patterned BC meta-skins. The non-infiltrative fibrous structure hosts a large volume of air inside, not only allowing the meta-skin to exhibit small elastic modulus and density but also leading to a huge acoustic impedance contrast with water, physically equivalent to an 'soft boundary'[22,23]. In this case, the paper-cutting BC meta-skin of deep-subwavelength thickness (viz., a patterned soft boundary) can be regarded as a new class of acoustic metasurfaces that can provide a sophisticated and full modulation of transmission acoustic field[22,24–28]. The ultrasound modulation in ~10 μm precision enables a wide spectrum of applications, such as high-quality acoustic holography[29], 3D ultrasonic tweezing[30,31], and perfect focusing[32,33], being especially significant in controlling the high-frequency (>5 MHz) ultrasound field's profile[34,35].

## Results

Meta-skin paper-cutting is meaningful in developing versatile ultrathin and super-light chip-scale ultrasonic devices with functionality customization. Here we create two prototypes through different paper-cutting patterns for the demonstration. The first one is a holey-structured meta-lens for generating high-quality ultrasound hologram (e.g., a letter 'H'). The holographic meta-lens operated at ~500 kHz has a thickness of only 20 μm (~$\lambda$/150), an area of $4 \times 4$ cm$^2$, and a weight of 18.6 mg (Supplementary Fig. 3). The second prototype is an imaging meta-lens for 3D reconstructions of a remote object in water via echoes of the focused ultrasound pulses (Supplementary Movies 5 and 6). The imaging meta-lens operates at 5 MHz, for which the diameter is 24 mm and the weight is only 10.3 mg (Supplementary Fig. 3). The focal FWHM (Full Width at Half Maximum) of the Fresnel meta-lens is ~195 μm and the focused pulse duration is ~220 ns, resulting in the far-field 3D echo imaging with the transverse resolution of only 190 μm and longitudinal resolution of 350 μm. The meta-skin paper-cutting technique permits versatile flexible functional devices for other advanced and adaptable technologies, such as ultrasonic diagnosis[36,37], non-destructive defective testing[38], and particle manipulation[39,40].

### Decorated BC meta-skin with superhydrophobic surface

The fabrication process of decorated BC meta-skin with super-hydrophobic surface is illustrated in Fig. 1a. BC nanofibers are hydrophilic in nature, which can be counter-intuitively transformed into a superhydrophobic material by judiciously incorporating polydimethylsiloxane-treated $SiO_2$ nanoparticles. However, owing to the big difference in surface energies between water and nano-particles, the large surface tension prohibits $SiO_2$ nanoparticles from entering into BC hydrogel, as shown in Fig. 1b. To solve this problem, we immerse BC hydrogel into ethanol solution with repeated ultrasonic treatment, and then transform it into BC alcogel. Since the BC alcogel possesses comparable surface energy with $SiO_2$ nanoparticles, we can disperse $SiO_2$ nanoparticles uniformly into the fibrous network of BC alcogel. In the ethanol phase, hydrogen bonds form between BC fibers and the treated $SiO_2$ nanoparticles, while the nanoparticles further agglomerate owing to the electrostatic force as well as the nanometer absorbability effect (Fig. 1c). Finally, we can obtain the $SiO_2$-nanoparticle-decorated BC meta-skin by removing the ethanol after the drying and hot-pressing process.

In Fig. 1d, e, we investigated surface and lateral morphologies of decorated BC meta-skin by scanning electron microscopy (SEM). Here the meta-skin exhibited a fibrous network structure bonded with chemically treated $SiO_2$ nanoparticles (diameter: ~50 nm), which is a porous structure that renders superhydrophobicity. In Fig. 1e, the sectional SEM image clearly shows that the sample thickness is only 20 μm. The porous structure with an ultrathin thickness makes the BC meta-skin ultralight (~121.5 mg for $4 \times 4$ cm$^2$) so that the sample can be readily resting on the dandelion's petals, as shown in Fig. 1f. The inset of Fig. 1f displayed the 'lotus effect' on the sample surface, where the CA of a water droplet reached up to 170.2° in the test.

The BC meta-skin is not only flexible but also has excellent machinability. By utilizing the laser-cutting technique, we can implement meta-skin paper-cutting to fabricate ultrathin and exquisite ultrasonic metasurfaces with custom-built functionalities, such as the holographic meta-lens for acoustic tweezing and the focusing meta-lens for 3D ultrasound imaging, shown in Fig. 2a. Moreover, the BC meta-skin can be regarded as bioactive metamaterials, for which any unwanted breakages can be repaired by bacteria under certain conditions, and the repaired BC meta-skin can be used in the paper-cutting devices for biomedical and ultrasound applications. Here we chose *G. xylinus* ferments to repair the damaged BC meta-skin. The bacteria-repairing process is shown in Fig. 2b, which basically contains three stages. In stage I, the BC meta-skin cut with a hole was immersed in the Hestrin-Schramm medium. In stage II, a thin BC layer was growing in the square hole via the secretion behavior of *G. xylinus* in the self-healing process. In stage III, the BC layer continued to grow thick enough to fully repair the hole. The BC meta-skin is also bio-degradable in the enzyme solution (Supplementary Fig. 4). For more details, please refer to Supplementary Notes 2 and 3.

To exhibit its excellent mechanical property, we cut a complex pattern on the BC-meta-skin via the focused laser beam, which comprises 'building', 'flag', 'HUST', and other elements (Fig. 2c), with very smooth edges and the narrowest line width ~200 μm (Fig. 2d). When the meta-skin was immersed in water, the sample was silver-flicking to the naked eyes, owing to the anomalous light reflection on the superhydrophobic surface (Fig. 2e). The stable air/water interface on the meta-skin provides a tremendous acoustic impedance mismatching for ultrasound and leads to a broadband total mirror reflection, as shown in Fig. 2f. We conducted the time-domain measurement (Fig. 2g) and spectral analysis (Fig. 2h) of reflection and transmission for a short ultrasound pulse at a central frequency of 5 MHz, which was normally incident on the meta-skin. Our results show that the incident pulse was reflected with an almost unchanged pulse shape with $\pi$-phase compensation, and the reflectivity reached 98.2% in broadband. A simplified model of BC meta-skin and numerical simulations of ultrasound reflection and transmission are appended in Supplementary Note 3, showing the important role of air/water interface in the total reflectance of ultrasound (Supplementary Figs. 5–7).

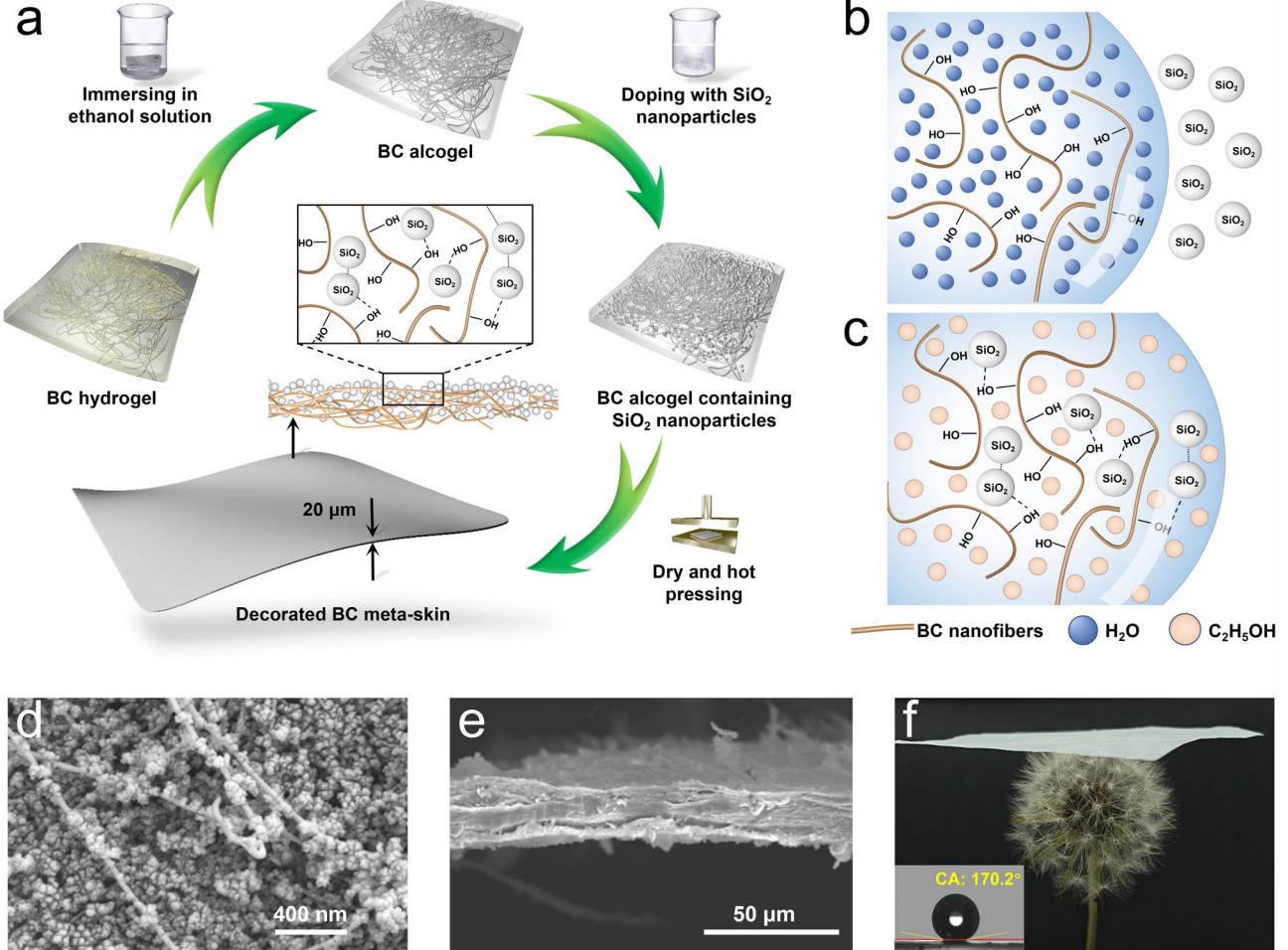

**Fig. 1 | Fabrication and characterization of decorated BC meta-skin.**
**a** Illustration of the fabrication process of the superhydrophobic BC meta-skin.
**b** Illustration of the hydrophilicity of BC nanofibers, where the hydrophobic SiO$_2$ nanoparticles can hardly enter into BC hydrogel. **c** Illustration of the ethanol-assisted method that allows SiO$_2$ nanoparticles to easily diffuse into the BC alcogel

and bond with BC fibers through intermolecular forces. **d**, **e** SEM images of the surface and sectional morphologies of the decorated BC meta-skin. **f** An optical image showing a BC meta-skin sample (4 × 4 cm$^2$) resting on dandelion's petals. The inset shows a strong '*lotus effect*' on the sample surface, with the CA of a water droplet being 170.2°.

The BC meta-skin paper-cutting provides a platform for realizing 2D functional ultrasonic devices, for example, the metasurface-based ultrasound holography. A meta-skin with through-holes can be regarded as a scatterer array for incident acoustic waves. Since the meta-skin thickness is much thinner compared with the side length, we can ignore the multiple-scattering process. A single-scattering process is typically described by the Fredholm integral equation, which can be deduced from the steady-state wave equation. The wave scattering problem with the Fredholm integral equation has been solved in quantum mechanics as the Lippmann-Schwinger equation. We discretize the meta-skin and divide it into a regular unit array to derive the discretization equation[41]. The acoustic pressure distribution on the metasurface can be further calculated from the nonlocal coupling between square holes at different positions,

$$p^{(I)} = \sum_J \left[\delta_{IJ} + jk\xi^{(I)}(1-\delta_{IJ})G(r^{(I)};r^{(J)})S_0\right]p^{(J)},\qquad(1)$$

$$G(r^{(I)};r^{(J)}) = \frac{e^{jk\cdot(r^{(J)}-r^{(I)})}}{4\pi|r^{(J)}-r^{(I)}|}.\qquad(2)$$

where $p^{(I)}$ is the pressure from the unit $I^{th}$, $S_0$ is the area of each unit, $G(r^{(I)};r^{(J)})$ is the Green's function between two units $r^{(I)}$ and $r^{(J)}$, $p^{(J)}$ is

the pressure from the unit $J^{th}$. $\delta_{IJ}$ is the Kronecker symbol, which indicates the self-radiation of unit $I^{th}$ when $I = J$. $\xi^{(I)}$ represents the characteristic of unit $I^{th}$. When unit $I^{th}$ is a through-hole unit, $\xi^{(I)} = 1$. When unit $I^{th}$ is a hard or soft boundary unit, $\xi^{(I)} = 0$. Next, we consider the continuity condition and obtain the transmission distribution of the acoustic pressure field on the meta-skin

$$p_t = \frac{1}{2}\left(E + S^{-1}\right)p_i,\qquad(3)$$

$$S^{(IJ)} = \left[\delta_{IJ} + jk\xi^{(I)}(1-\delta_{IJ})G(r^{(I)};r^{(J)})S_0\right].\qquad(4)$$

Here, $p_t$ is the transmitted acoustic pressure, $p_i$ is the incident acoustic pressure, $E$ is the identity matrix, $S$ is the scattering matrix originating from Eq. (1) and $S^{(IJ)}$ is the matrix element of $S$. The radiated pressure field out of the plane can thus be established based on the on-plane pressure field, by regarding the on-plane pressure field as a new acoustic source. Hence, it is feasible to obtain a pre-designed acoustic field in the far field by adjusting the arrangement of units on the meta-skin. Here we label the through-hole unit as '1' and the soft-boundary unit as '0'. Therefore, the question of obtaining the target acoustic field is attributed to a '0–1' programming on the discretized meta-skin. It is effective to solve the '0–1' programming via the optimization

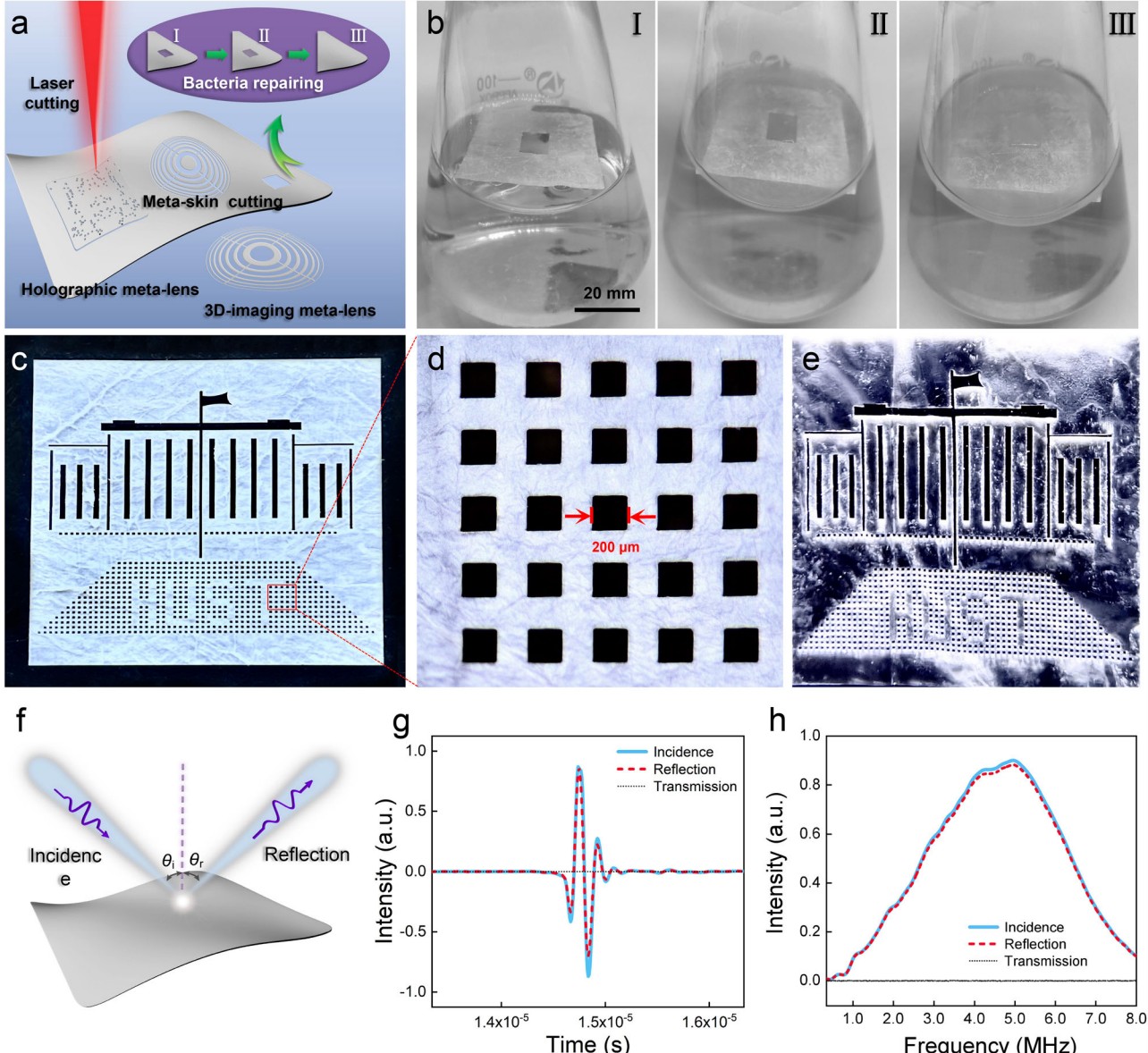

**Fig. 2 | Mechanical, acoustic, bacteria-repairing properties of the BC meta-skin.**
**a** Illustration of BC meta-skin paper-cutting, via which we can realize the ultrathin meta-lenses with different functionalities. The inset shows that the BC meta-skin has a surprising bacteria-repairing property. **b** Three stages (I, II, III) in the bacteria repairing of a broken meta-skin. **c** Complex pattern comprising 'building', 'flag', 'HUST', and other elements. **d** Laser-cut BC meta-skin with the narrowest line width of ~200 μm, showing excellent processability. **e** Optical image of the patterned meta-skin immersed in water, where the silvery surface indicates the existence of a very stable air-filled interface (or the Cassie-Baxter states) on the meta-skin. **f** Illustration of perfect mirror reflection for ultrasound on the nanosurface due to the stark acoustic impedance mismatch. **g** The time-domain measurement in reflection (with $\pi$-phase compensation) and transmission for a short ultrasound pulse incident on the meta-skin. **h** Spectral analyses of incident, transmitted, and reflected signals, indicating broadband ultrasound reflection.

method[41,42]. Details of the optimization algorithm are given in Supplementary Note 4.

The illustration of an acoustic hologram based on meta-skin paper-cutting is shown in Fig. 3a, where a plane-wave ultrasound at 500 kHz is finely modulated by a holey-structured meta-skin to project a hologram in the far field. The sample fabricated by laser cutting is shown in Fig. 3b, for which the side length is 30 mm, the thickness is 20 μm, the size of one through-hole unit is $300 \times 300$ μm², and the weight is 18.6 mg. In our holography experiment, the finely modulated ultrasound beam by the sample mask was projected to form a hologram pattern 'H' at 35 mm behind the lens. In Fig. 3c, the theoretical, simulation and measured results are presented with 'H'-like distributions. Compared with the theoretical result, the mean square errors (MSEs) of the simulation and experimental results are 0.1294 and 0.1425,

respectively. In Fig. 3d, we compare the intensity distributions on section lines I, II and III (as marked by the white dashed lines in Fig. 3c) quantitatively, where the results manifest a close resemblance. It should be mentioned that in the theoretical calculation, we focus on the case in free space without considering boundary reflection. The experimental measurement was carried out in a water tank ($70 \times 120 \times 80$ cm³; Supplementary Fig. 8). Since the water tank is large enough, the wave propagation can be regarded as in free space. In the 3D simulation, the model cannot be set very large due to the limitation of computation resources. Therefore, the plane-wave radiation boundary condition cannot provide a perfect absorption for obliquely incident waves, resulting in unwanted side peaks.

The meta-skin paper-cutting can also be utilized to realize ultrathin and ultralight 3D imaging meta-lenses. Here the meta-skin imaging

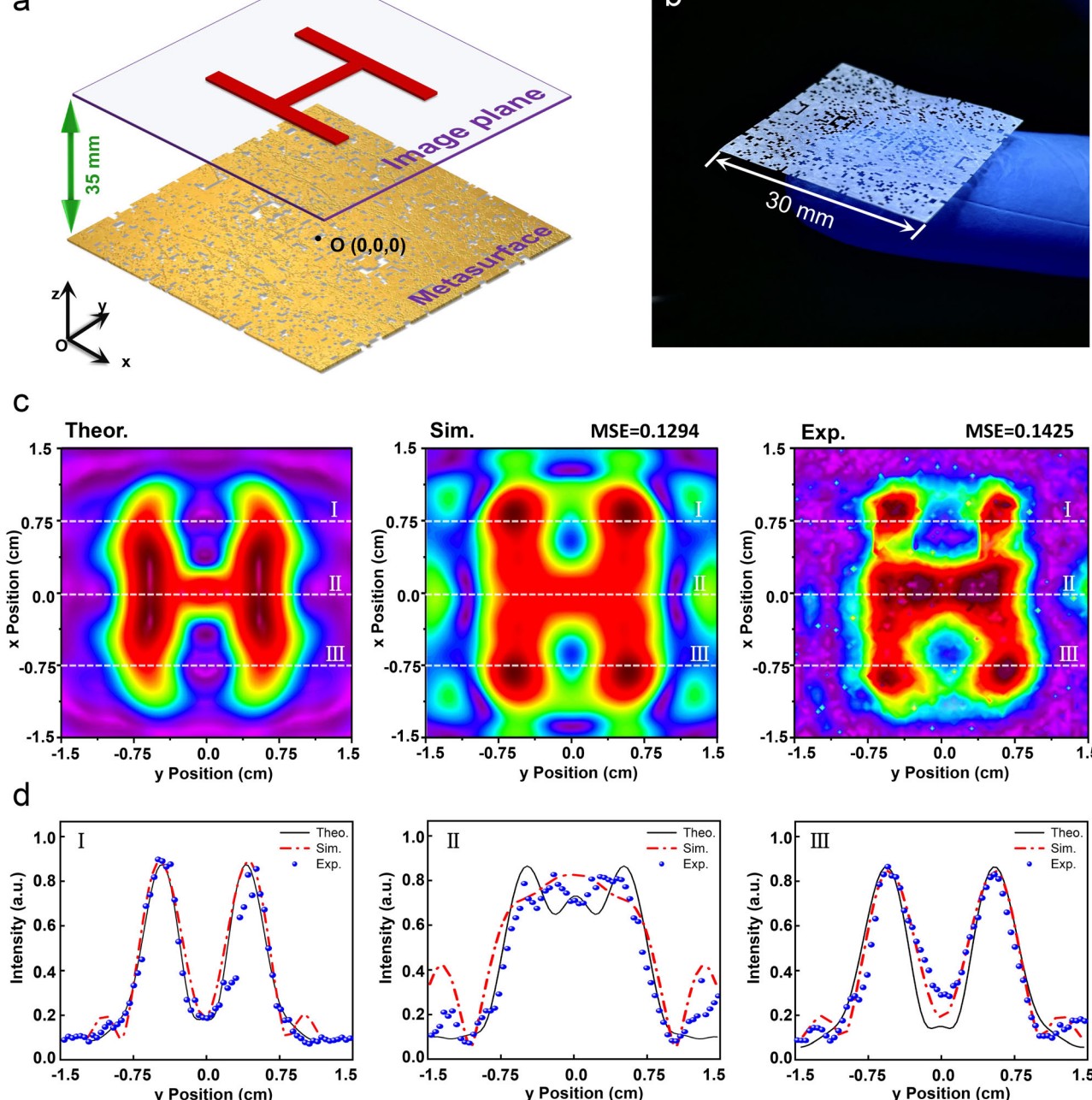

**Fig. 3 | A holographic meta-lens via the BC meta-skin paper-cutting.**
**a** Illustration of the ultrasonic hologram via BC meta-skin paper-cutting. The distance between the holographic meta-lens and the hologram plane is 35 mm. **b** An optical image showing the ultrathin and ultralight holographic meta-lens fabricated via BC meta-skin paper-cutting. **c** Intensity fields on the hologram planes from the calculated, simulated and experimentally measured results. **d** Intensity distribution on section lines I, II and III, as marked by the white dashed lines in (**c**) ($x = 0.75$cm, $0.0$cm, $-0.75$cm, respectively) for quantitative comparison.

lens has 18 concentric rings of the radii $r_n (n = 1, 2, 3 \ldots)$, where $r_n$ satisfies the following relation[43]

$$r_n = \left[ \left( f + \frac{n\lambda}{2} \right)^2 - f^2 \right]^{1/2} \quad (n = 1, 2, 3 \ldots), \quad (5)$$

where $\lambda$ is the ultrasound wavelength and $f$ is the focal length. The width of each circular strip is

$$w_n = r_{n+1} - r_n \quad (n = 1, 2, 3 \ldots), \quad (6)$$

where $w_n$ is the width of the $n$th strip numbered from the center, and the detailed design is shown in Supplementary Fig. 9. The sample is shown in the inset of Fig. 4a, for which the diameter is 24 mm, the thickness is 20 μm, the narrowest width of the circular strip is 240 μm, and the weight is 10.3 mg. The source is a planar signal emitter and probe, with an operation frequency of 5 MHz and a duration of ultrasonic pulse of ~980 ns. For the paper-cutting meta-lens, FWHMs of focal spots in $x$-$y$ plane and in $z$-axis are $0.63\lambda$ (~190 μm) and $6\lambda$ (~1800 μm), respectively (Supplementary Fig. 10). The 3D ultrasonic image is reconstructed by using the Hilbert transform to the probed echo pulses, with the lateral and longitudinal resolutions determined by the

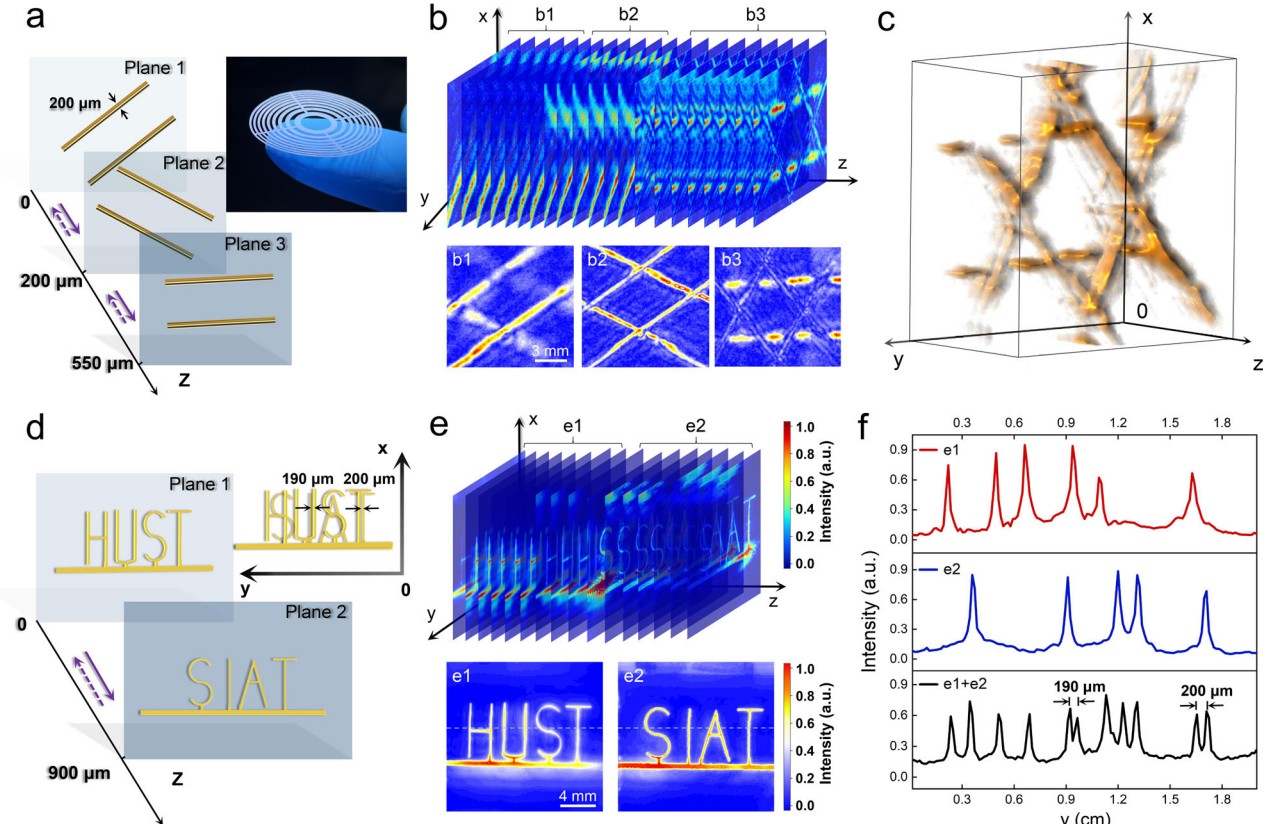

**Fig. 4 | A 3D imaging meta-lens via BC meta-skin paper-cutting. a** Graphite rods for 3D ultrasonic imaging, for which the diameter is 200 μm. The graphite rods are set to position in three planes (1, 2, 3) with different orientations for testing the longitudinal resolution, and the projection forms a hexagram pattern. Inset shows an imaging meta-lens based on BC meta-skin paper-cutting. **b** The ultrasonic images of graphite rods with different depths (b1, b2, b3). Here the 3D ultrasonic images are reconstructed by extracting and arranging the echo signal data. **c** 3D reconstruction image of graphite rods. **d** 3D-printed metal patterns 'HUST' and 'SIAT' for 3D ultrasonic imaging. The distance between the two patterns is 900 μm. The horizontal shift between letters 'T' is 200 μm for testing the transverse resolution. **e** Reconstructed images of 'HUST' and 'SIAT' at the depths of e1, e2. **f** Intensity profiles extracted on the dashed lines (e1, e2) in (**e**). We also show the superimposed result of e1+e2.

spot size of focused ultrasound in *x-y* plane and the pulse duration, respectively, as shown in Supplementary Notes 6, 7.

For the imaging experiments, we first utilized six graphite rods (diameter: 200 μm) that are positioned on three planes ($z$ = 0, 200, 550 μm) to test the longitudinal resolution of the imaging set-up, as shown in Fig. 4a. The three planes are all covered by the focal spot of meta-lens in *z*-axis. Therefore, 3D imaging of graphite rods can be reconstructed by scanning only in *x* and *y* directions and extracting the echo pulses. In Fig. 4b, we present the 3D volume imaging data of multiple slices, where the source data are provided in the source data file, and each slice contains the object information in different depths along *z*-axis. The slices can be divided into three clusters b1, b2, and b3, corresponding to the images of graphite rods in planes 1, 2, and 3, respectively. From Fig. 4b, the superimposed images of slice clusters b1, b2, and b3 clearly show that there exists an artifact of b1 in the superimposed image of b2, indicating that the proposed set-up cannot distinguish two objects separated by 200 μm in *z* direction. However, the superimposed image of b3 has no artifacts of b1 or b2, showing that the set-up reaches the longitudinal resolution of 350 μm. We note that the image of b3 is disconnected, which is caused by missing the echo signals from plane 3 due to the object blocking in planes 1 and 2. Figure 4c shows the 3D reconstruction image of graphite rods, forming a hexagram pattern in the angle of view.

To determine the transverse resolution of the proposed BC metasurface, we used the metal-printed 'HUST' and 'SIAT' patterns as the imaging objects. The two objects were separated by 900 μm along *z*-axis, which was within the coverage of the needle-like focal spot. To test the resolution, two letters 'T' of 'HUST' and 'SIAT' were offset by 200 μm in *y* direction, while the letters 'U' and 'I' were offset by 190 μm, as shown in Fig. 4d. The experimental results are presented in Fig. 4e, for which the information of 'HUST' and 'SIAT' can be extracted from the slice clusters e1, e2. The superimposed images of e1 and e2 in Fig. 4e reveal that two objects spacing 900 μm can be well distinguished without the interference of artifacts. In Fig. 4f, we show the intensity profiles along the dashed lines in e1 and e2 (the red and blue lines) and present the superposition result of e1+e2 (the black line), which demonstrates that the offsets of 190 μm and 200 μm can be clearly distinguished. Above all, from Fig. 4, we clearly demonstrate that the longitudinal and lateral resolutions of the proposed BC metasurface lens are 1.16$\lambda$ (~350 μm) and 0.63$\lambda$ (~190 μm) at 5 MHz, respectively.

## Discussion

In conclusion, we report the decorated BC meta-skin combined with the paper-cutting for the realization of ultrathin and ultralight chip-scale ultrasonic devices of versatile functionalities. We demonstrate that the proposed BC meta-skin is very stable in the water environment, which remains superhydrophobic in water for over 200 days. The excellent machining property with only 20 μm thickness makes it convenient for high-precision paper-cutting via the laser-cutting technique. The BC meta-skin is also a bioactive metamaterial, which has the interesting property of bacteria repairing the breakages. Due to the micro-cavities in BC meta-skin decorated with hydrophobic silica

nanoparticles, a stable and robust air-filled interface is formed, which provides an ideal soft boundary for ultrasound with total wave reflection. The BC meta-skin paper-cutting enables a high-accuracy (-10 μm) full amplitude modulation of ultrasound beams. We successively realized two ultralight (<20 mg) chip-scale ultrasonic devices, viz., holographic meta-lens and 3D imaging meta-lens. Nonlocal couplings in the holey-structured meta-lens result in acoustic holograms, which is meaningful for realizing the holographic ultrasonic tweezers. The concentric-slit structured ultrasonic meta-lens can be employed for realizing high-resolution 3D echo imaging of far-field objects. Our approach is compatible with high-frequency functional 2D devices and origami/kirigami techniques, which is significant for promoting the applications of nano/micro ultrasonic metamaterials in advanced biomedical engineering technologies.

## Methods

Figures 1a and 2a were generated by the software C4D (Cinema 4D R24) and PPT (PowerPoint2021). The extraction and processing of experimental data were done by Matlab 2021a and Origin 2021. The simulation data were completed by Comsol Multiphysics 5.6.

### Fabrication and characterization of BC meta-skins

The BC wet membranes were purchased from Hainan Yide Foods Co. Ltd. (China), and the average full swelling thickness was 2 mm. SiO$_2$ nanoparticles were purchased from Evonik Degussa Co., and the average particle size was 50 nm. Ethanol absolute was purchased from Shanghai Sinopharm Chemical Reagent Co., Ltd. We dispersed 2.5 w% SiO$_2$ nanoparticles into the ethanol solution, which was treated by ultrasound for 0.5 h to obtain the SiO$_2$ nanoparticles dispersing solution. We stored it at 2–8 °C for further use.

The BC wet membranes were first cut into a square shape with a side length of 10 cm, which was treated in boiling water until full swelling. Then, the swollen BC hydrogels were repeatedly immersed in the ethanol solution with ultrasound for 1 h and shaken for 6 h until we obtained the BC alcogels. The BC alcogels were submerged into the SiO$_2$ nanoparticles dispersing solution. Through ultrasound and shaking, the SiO$_2$ nanoparticles were fully dispersed into the 3D fiber network structure. The BC alcogels containing SiO$_2$ nanoparticles were dried naturally and washed to wipe off undecorated SiO$_2$ nanoparticles. Finally, we used the hot-pressing to make it into a thin and flat superhydrophobic BC meta-skin.

The morphologies of the BC meta-skins were characterized by utilizing the Nova NanoSEM 450 field emission scanning electron microscope (SEM). The contact angle was measured by utilizing the OCA20 optics contact angle meter. The measurement results in Supplementary Note 1 show that the tensile strengths and Young's modulus become slightly larger with the increase of SiO$_2$ concentration. For example, the average tensile strengths of the hydrophilic BC membrane (0% SiO$_2$ concentration) and hydrophobic BC meta-skin (2.5% SiO$_2$ concentration) were measured to be 52.8 MPa and 59.9 MPa, respectively. In the tensile failure test, the maximum stresses undertaken by the same-size BC membrane and BC meta-skin are 65.1 MPa and 77.7 MPa, respectively. The contact angle of meta-skin also becomes larger with the increase of SiO$_2$ concentration, for which the contact angle is stable at around 160° when the concentration is above 0.5%, indicating the existence of superhydrophobicity.

### Preparation of acoustic meta-lens

The patterned meta-lenses for different application scenarios were designed via modeling and simulation. The patterns were realized via paper-cutting by using the laser to process the BC meta-skin. The paper-cutting has high precision due to the excellent mechanical properties of BC meta-skin. The fabricated acoustic meta-lenses were stable in water, for which the surface kept superhydrophobic

during the immersion in water for more than 200 days. Therefore, the processed BC meta-skin lens is compatible with commercial uses. With damages, the meta-skin can be repaired through bio-active self-healing.

### Holographic meta-lens and Imaging meta-lens

Here we chose 500 kHz and 5 MHz for ultrasound holography and 3D imaging to demonstrate that our meta-skin lens can modulate ultrasound in broadband. As for the imaging meta-skin lens, we chose the frequency of 5 MHz to achieve a higher image resolution. For the holographic meta-skin lens, we chose the frequency of 500 kHz for achieving a hologram field pattern on a large scale and further benefited the ultrasonic tweezing of large objects.

The holographic meta-lens comprises regularly arranged square holes with a side length of 300 μm (1/10 wavelength of ultrasound at 500 kHz), and the thickness is only 20 μm for 1/150 of the wavelength. The acoustic scattering properties are derived from the nonlocal coupling between square holes at different positions. Therefore, meta-skin holography can be regarded as reshaping the propagation behavior of ultrasound through the interaction between acoustic waves themselves rather than the interaction between acoustic waves and metamaterial with an interior space[41]. In the incident ultrasound wave, the reflected and transmitted waves interact and change their wave fields synergistically through the nonlocal effects between the arranged through-holes.

The required distribution of through-holes for targeted hologram was optimized by using the commercial software MATLAB through the genetic algorithms. The computer cluster used for calculations and simulations was configured with a GeForce RTX 3090 Graphics Processing Unit (GPU), Intel(R) Xeon(R) CPU Gold 6226R @ 2.40 GHz and 1.5 T RAM. The source used in the experiment (OLYMPUS IL0.512GP Diameter = 3.8 cm) emitted single-frequency ultrasound at the frequency of 500 kHz.

The imaging meta-lens comprises concentric slits which can focus ultrasound into a needle-like spot in the far field. For echo imaging, a portion of ultrasound energy in the needle-like spot is reflected back by the objects at different depths, for which a series of pulse-wave packets can be observed in sequence in the time domain of the echo signal (Supplementary Fig. 11). In our experiment, the first wave packet in the time domain of the echo signal comes from the meta-lens. Therefore, when processing the time-domain signal, we must filter out the first wave packet. We demodulate the echo signal $s(t)$ to obtain the profile $\hat{s}(t)$ through Hilbert transformation

$$\hat{s}(t) = H[s(t)] = \int_{-\infty}^{\infty} \frac{s(\tau)}{t - \tau} d\tau, \tag{7}$$

where 'H[ ]' represents the Hilbert transform of the signal and $\tau$ is the arrival time. By 2D sweeping the field via the needle-like spot, we can readily obtain the 3D imaging data (Supplementary Figs. 12 and 13), which comprise different slice data that correspond to different time $t$ as well as the information of objects at different depths.

In the imaging experiment, the pulse transmitter and receiver (DPR-300) machine was used to transmit and receive ultrasonic pulses. The loading voltage was 200 V, the transducer (Olympus-5 MHz) adopted the echo mode, and the pulse repetition frequency was 1 kHz (Supplementary Fig. 14). The sampling frequency of the acquisition card was 100 MHz, and an average of 16 times was taken. Data were processed by using the software MATLAB. The computer cluster used for calculations and simulations was configured with GeForce RTX 3090 graphics processing units (GPUs), Intel(R) Xeon(R) CPU Gold 6226R @ 2.40 GHz, and 1.5 T RAM.

## Data availability

The authors confirm that all relevant data are included in the paper and/or its supplementary information files, and the raw data are available upon request from the corresponding author. Source data are provided with this paper.

## Code availability

Codes related to experiments in the main text are placed in Supplementary Information (Source Code).

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

## Acknowledgements

This work was supported by the National Key R&D Program of China (Grant Nos. 2020YFA0211400 to X.-F.Z., 2020YFA0211401 to X.-F.Z., and 2020YFA0908800 to F.L.), the National Natural Science Foundation of China (Grant Nos. 51973076 to G.Y., 52373235 to G.Y. and 12074402 to F.L.), the Shenzhen Basic Research Program (No. JCYJ2020010911 4825064 to F.L.), the Shenzhen Science and Technology Program (No. GJHZ20210705141404012 to F.L.), Natural Science Foundation of Guangdong Province (No. 2023B1515040008 to F.L.), and the Key Laboratory for Magnetic Resonance and Multimodality Imaging of Guangdong Province (No. 2023B1212060052 to F.L. and H.-R.Z.).

## Author contributions

X.-F.Z. conceived the concept. Z.-L.L., K.C., Z.-J.S., P.-Q.L. and Y.-G.P. fabricated and characterized the sample. Z.-L.L., L.-X.H. and F.L.

completed the 3D ultrasonic imaging experiments. Z.-L.L. and Q.-L.S. completed the acoustic hologram experiments. Z.-L.L., K.C. and X.-F.Z. wrote the manuscript. X.-F. Z, G. Y. and H.-R. Z. supervised the whole project.

## Competing interests

The authors declare no competing interests.
