## [Peer Review File · Nature Communications]

Decorated bacteria-cellulose ultrasonic metasurfaceReviewer #1 (Remarks to the Author):

This submission by Li et al. presents a study on a decorated bacteria-cellulose acoustic metasurface for ultrasonic applications. The use of SiO₂ nanoparticles in the bacteria-cellulose metasurface helps to achieve superhydrophobic properties as well as stability. The authors also showed the proposed metasurface has the ability of self-repair because of its bioactive nature. Finally, by integrating with laser cutting, the metasurface is demonstrated to be capable of achieving acoustic holography and imaging.

Metasurfaces have been an active topic in recent years, and extending their usage in biomedical applications in a biocompatible manner is an important direction. This work reports a feasible approach to achieve this goal, and the results are supported by experimental data. That said, I would like to support its publication given the following points are carefully addressed.

(1) The proposed meta-skin exhibits a fibrous network with embedded SiO₂ nanoparticles. It seems that the mechanical properties are mainly contributed by the cellulose as well as the SiO₂ nanoparticles, please clarify if that is the case. Along this line, it is unclear how bacteria would be important in certain biomedical or ultrasound applications. It will be good to give some context on that.

(2) Are the bacteria active for all the proposed meta-skin? If so, is special treatment needed to keep them active (e.g., in the case shown in Supplementary Fig. 1b)? Another concern is about the self-repair capability. While bacteria-cellulose metasurface exhibits this property, the required kirigami pattern for a certain application would be destroyed after the repair process, right? Please clarify.

(3) Fig. 2e is confusing. Did the authors overlay the signals? The three signals should be separated in the time domain. Also for reflection the signal should be reversed in time.

(4) Since the hologram is demonstrated at 500 kHz, the spectral analysis in Fig. 2f should cover this range. No data is shown at 500 kHz.

(5) From the videos it seems that the meta-skin is a bit tilted in the corners. I assume the authors use a planar condition when calculating the properties for imaging and hologram, and inaccuracies may arise from a non-negligible tilting. Did the authors do anything about that in the acoustic measurements?

(6) Ref [41] has been published, see Li, Xin, et al. "Ultrathin Acoustic Holography." *Advanced Materials Interfaces* (2023): 2300034. Please update.

(7) There are a few typos in the manuscript, e.g., in line 102 to enter BC hydrogel; line 175 feasible to use.

Reviewer #2 (Remarks to the Author):

The authors present a study on meta-skin made from SiO₂-nanoparticle decorated bacteria-cellulose, which exhibits unprecedented properties such as ultra-lightness, stability in water, small moduli, and high impedance. These properties enable the material to achieve various functionalities, including holography, focusing, and imaging. This work presents an interdisciplinary area among physics, biology, environment, and engineering. I was attracted by the manuscript, but there are several areas that the authors may consider to improve so that their manuscript is more understandable to a general audience.

1. In the first paragraph, the authors mentioned the current challenges in the fabrication of superhydrophobic interfaces and proposed solutions. However, the link between the solutions and the challenges is not quite clear. To make their contribution clearer in terms of tackling the challenges, the authors can elaborate more on how their proposed solutions address the challenges.

2. Including a theoretical analysis of the transverse and longitudinal resolutions for the meta-lens may provide readers with a better understanding of the meta-lens's performance.

3. The authors may consider to include a model to characterize the achieved meta-skin, for example, its acoustic parameters. Such a model can help readers understand the meta-skin's behavior and its potential applications.

4. The authors should explain the abbreviation FWHM, which is not immediately clear to readers.
5. In the supplemental video, the image of the line under SIAT appears from left to right, while the image of the line under HUST appears from the middle to two sides. Is there any reason for these two different behaviors?
6. The authors designed the holography and meta-lens for different frequencies, i.e., one is 500kHz and another is 5MHz. The authors may explain why they chose different frequencies and how this affects the performance of the holography and meta-lens.
7. In Fig 3 d-II, the simulation results show a single peak in the center without two side peaks, which is quite different from the theory and the experiment. The authors should explain this discrepancy.

Reviewer #3 (Remarks to the Author):

The paper presents a novel mechanism for transforming hydrophilic Bacterial Cellulose (BC) into a hydrophobic structure by carefully treating BC nanofibers with polydimethylsiloxane-coated SiO₂ nanoparticles. The authors claim that their process, which involves converting the BC hydrogel into a BC alcogel and subsequently treating it with SiO₂ nanoparticles, is a unique approach. This treatment results in a meta-skin with enhanced flexibility and machinability, enabling the creation of kirigami structures through laser cutting. Although I am not an expert in the specific fabrication process, I find the description to be both intriguing and clear. The proposed method has the potential to attract significant interest from researchers working on acoustic metamaterials and fabrication, who may further explore the design and application of these materials.

The presentation of results in Figure 2 is somewhat confusing. Figures 2a and 2g appear to be related, while Figures 2d, 2e, and 2f pertain to the same measurement, and Figures 2b and 2c form a third group of data. To improve clarity, the figures and their corresponding descriptions should be reorganized to present a more coherent set of results. For instance, Figures 2a and 2g could be grouped together, along with the relevant text that discusses these figures.

Additionally, to enhance the comprehensibility of Figure 3a, it would be beneficial to include axis labels, as seen in Figure 3c, and provide a clear indication of the origin (0,0,0).

I find it unclear why the authors describe their structure as a "kirigami" structure. While the treated BC undergoes laser cutting to create a pattern of holes that functions as a transmissive meta-structure, I do not observe any folding in the design. In traditional kirigami structures, folding is typically intentional, incorporated during fabrication to achieve specific properties or functionalities. Thus, the term "kirigami" may not accurately represent the surface in this case.

I disagree with the authors' claim of good agreement between the theoretical, simulation, and experimental results. The presented data is not conclusive, as it relies on visual comparison, which is subject to various uncertainties, such as print resolution and choice of scale and colors. I suggest presenting the data as a 2D image (instead of the 3D perspective view shown in the paper) and employing similarity metrics like SSIM or MSE for comparison, rather than relying solely on visual inspection. While visual comparison is relevant and useful, additional metrics would strengthen the analysis.

The core argument and contribution of this paper seem to revolve around the creation of bio-active, superhydrophobic BC structures. Although using through-hole patterning to manipulate sound waves is not a novel concept, the paper demonstrates the potential of BC meta-skin for acoustic holography and meta-lenses, thereby providing proof of concept for its effectiveness in manipulating sound waves. I recommend that the authors either refrain from using the term "kirigami" or provide a clearer explanation for why their structure genuinely incorporates both folds and cuts.

Finally, I also feel that the paper is missing some key intermediate results/ characterisations. Investigating the ability to manipulate the density of SiO₂ nanoparticles in the BC hydrogel can be beneficial for understanding and tuning various properties of the material. Some studies should be conducted to understand and characterise this better. For example, what is the distribution and uniformity of the nanoparticles in the BC hydrogel?

It would be good to know more about the mechanical properties of the BC meta-skin (tensile strength, elastic modulus, and ductility) and how it changes with different SiO₂ density.

The contribution of this paper is both novel and interesting. I feel it would make a nice contribution to Nature Comms. But I would also appreciate more rigorous testing and characterisation of the properties of the meta-skin before it can be included for publication.

Response Letter to Reviewers

We are grateful for the insightful and constructive comments from all the reviewers which are really helpful to improve the quality of our manuscript. In the text below, each of the comments from each reviewer is quoted in italics (light blue) and is followed by the corresponding detailed response. We also revised the manuscript and added the Supplementary Materials accordingly, and these changes are highlighted in those files.

Reply to the Reviewer #1

Main comments:

This submission by Li et al. presents a study on a decorated bacteria-cellulose acoustic metasurface for ultrasonic applications. The use of SiO₂ nanoparticles in the bacteria-cellulose metasurface helps to achieve superhydrophobic properties as well as stability. The authors also showed the proposed metasurface has the ability of self-repair because of its bioactive nature. Finally, by integrating with laser cutting, the metasurface is demonstrated to be capable of achieving acoustic holography and imaging.

Metasurfaces have been an active topic in recent years, and extending their usage in biomedical applications in a biocompatible manner is an important direction. This work reports a feasible approach to achieve this goal, and the results are supported by experimental data. That said, I would like to support its publication given the following points are carefully addressed.

Response: Thank you for your careful review and constructive suggestions. In the following, we will try our best to answer reviewer's comments.

Minor comments:

1. The proposed meta-skin exhibits a fibrous network with embedded SiO₂ nanoparticles. It seems that the mechanical properties are mainly contributed by the cellulose as well as the SiO₂ nanoparticles, please clarify if that is the case. Along this line, it is unclear how bacteria would be important in certain biomedical or ultrasound

applications. It will be good to give some context on that.

Response:

We sincerely appreciate the valuable suggestions provided by the reviewer. The mechanical properties are mainly contributed by cellulose, while the doping of SiO₂ nanoparticles also enhances the mechanical properties slightly. Nanofibers in bacterial cellulose intertwine with each other through physical and chemical binding, enhancing the tensile strength of bacterial cellulose membrane. To deeply analysis the mechanical properties of the SiO₂-decorated membrane, we conducted the test to gain the tensile strength and Young's modulus and the results are showed in Fig. R1. During the experiments, the samples were prepared with dispersing solution under different SiO₂ concentration of 0% (Pure BC), 0.05%, 0.1%, 0.5%, 1% and 2.5% (BC meta-skin). The strain-stress curves of BC membrane and BC meta-skin are displayed in Fig. R2. As Fig. R1 exhibits, the average tensile strength of pure BC membrane is 52.8 MPa and that of BC meta-skin is 59.9 MPa. Meanwhile, it is obvious that both of the tensile strength and Young's modulus increase slightly with the concentration of SiO₂ particles increases. SiO₂ particles act as a lubricant when they get into the fibrous BC network, resulting in the improvement of sliding properties between nanofibers and providing a strong shear force which augments the tensile strength and Young's modulus.

In this study, the basic raw BC membrane is secreted by bacteria (*G. xylinum*) and the repairing process also needs the bacteria to act as a dressmaker to repair the damage of BC meta skin. Bacteria (*G. xylinum*) play an important role in this search. It is the secretory behavior of bacteria that endows the BC meta-skin with excellent mechanical properties, self-repairing, degradability, fibrous network structure, and so on. In the self-repairing process, bacteria in the damaged part secrete newborn bacterial cellulose to repair the unwanted breakage, and the repaired devices can be used for biomedical and ultrasound applications.

We added Figs. R1 and R2 as Figs. S1(e) and S1(f) and made some discussions in the Supplementary Note 1. Property and stability of BC meta-skin (Pages 3-5). We also give contexts on how bacteria would be important in certain biomedical or ultrasound applications in main text (Page 7 lines 142-145).

Fig. R1. The tensile strength and Young's modulus of SiO₂-nanoparticle decorated BC membrane prepared under dispersions with different SiO₂-nanoparticle concentrations.

Fig. R2. The strain-stress curves of the pure BC membrane and decorated BC meta-skin.

2. Are the bacteria active for all the proposed meta-skin? If so, is special treatment needed to keep them active (e.g., in the case shown in Supplementary Fig. 1b)? Another concern is about the self-repair capability. While bacteria-cellulose metasurface exhibits this property, the required kirigami pattern for a certain application would be destroyed after the repair process, right? Please clarify.

Response: We are grateful for the questions raised by the reviewer. In our work, the bacteria (*G. xylinum*) were wiped out in the final product of BC meta-skin. Specifically, Fig. 1b of the main text showcases that the BC hydrogel contains a plenty of hydroxyl groups and water, preventing the entrance of superhydrophobic SiO₂ nanoparticles. Here, the BC hydrogel is secreted by bacteria and has gone through the treatments of

boiling 1% NaOH solution and the pure water to remove bacteria and metabolite. No bacterium will be left on BC hydrogel and therefore it is unnecessary to conduct special treatment to keep bacteria active. In this work, we only introduced active bacteria for secreting the BC, for example when fabricating bacterial cellulose as the raw material to prepare BC meta-skins or in the self-repairing process.

As reviewer questioned, we focused on repairing the unwanted damages during the self-repairing process, which may include the useful paper-cutting patterns. However, it is feasible and effortless to conduct the paper-cutting process on the self-repaired BC meta-skin for further usage in biomedical and ultrasound applications. Please refer to the revised main text (Page 7 lines 142-145).

3. Fig. 2e is confusing. Did the authors overlay the signals? The three signals should be separated in the time domain. Also for reflection the signal should be reversed in time.

Response:

We thank the reviewer for the good questions. In revised supplementary materials, we show the acoustic setup for measuring the incident, transmitted and reflected signals.

Fig. R3. Acoustic setup for measuring the incident, transmitted and reflected signals.

Here we have measured the incident and reflected pulses of ultrasound on the meta-skin at the central frequency of 0.5 MHz. The acoustic setup is shown in Fig. R3(a). The ultrasound pulse was launched by the bottom transducer. For measuring incident

pulse, the ultrasound signal was received by the top transducer directly without setting the meta-skin in the propagation path. For measuring the reflected and transmitted pulses, ultrasound signals were received by the bottom and top transducers respectively by setting the BC meta-skin in the propagation path, where the received signals were actually reflected from or transmitted through the meta-skin surface. Here we set $h_1=h_2$ to make sure the incident, reflected and transmitted pulses are in the same time slot. The reflected pulse was compensated by π phase due to the soft-boundary-like surface of BC meta-skin for comparison with the incident one. In the revised supplementary, we added the related information on measuring and processing signals (Supplementary Fig. 7 Pages 14-15).

4. Since the hologram is demonstrated at 500 kHz, the spectral analysis in Fig. 2f should cover this range. No data is shown at 500 kHz.

Response: We thank the reviewer for the good questions. In the revised main text, the spectrum in Fig. 2(h) covers the range at 500 kHz (or 0.5 MHz). Since the central frequency of the transducer used in Fig. 2(h) is 5 MHz, we also add the spectrum data for the transducer at the central frequency of 0.5 MHz in the supplementary materials (Supplementary Fig. 7 Pages 14-15). In Fig. R4(a), we performed the time-domain measurements for both incident and reflected pulses, where the reflected pulse was compensated by π phase for comparison with the incident one. In Fig. R4(b), the spectral analyses show that incident ultrasound pulse has been completely reflected in broadband (0.1~1 MHz).

Fig. R4. **a**, The time-domain measurement of incident, reflected, and transmitted pulses with a center frequency at 0.5 MHz. **b**, The spectral analyses of incident, reflected, and transmitted signals.

5. *From the videos it seems that the meta-skin is a bit tilted in the corners. I assume the authors use a planar condition when calculating the properties for imaging and hologram, and inaccuracies may arise from a non-negligible tilting. Did the authors do anything about that in the acoustic measurements?*

Response: We thank the reviewer for this good question. Yes, it can be seen from the Supplementary Video 4 that the corners of the meta-skin are slightly tilted, and this tilt is caused by the ultra-thin nature of BC meta-skin. Therefore, in the experiment, we fixed the meta-skin lenses on a cling film of ultrasound transparency, as shown in the Supplementary figures 8, 12, and 13.

6. *Ref [41] has been published, see Li, Xin, et al. "Ultrathin Acoustic Holography." *Advanced Materials Interfaces* (2023): 2300034. Please update.*

Response: We thank the reviewer for alerting us to this paper citation. This paper has updated citations in the revised text

[41]. Li, Xin, et al. Ultrathin Acoustic Holography. *Adv. Mater. Interfaces* 10.8 (2023): 2300034.

7. There are a few typos in the manuscript, e.g., in line 102 to enter BC hydrogel; line 175 feasible to use.

Response: We thank the reviewer for bringing this problem to our attention. We have revised this problem in the latest manuscript submission. The other parts of the text are also checked in details.

Reviewer #2 (Remarks to the Author):*Main comments:*

The authors present a study on meta-skin made from SiO₂-nanoparticle decorated bacteria-cellulose, which exhibits unprecedented properties such as ultra-lightness, stability in water, small moduli, and high impedance. These properties enable the material to achieve various functionalities, including holography, focusing, and imaging. This work presents an interdisciplinary area among physics, biology, environment, and engineering. I was attracted by the manuscript, but there are several areas that the authors may consider to improve so that their manuscript is more understandable to a general audience.

Response: We are very grateful to this reviewer for the high-standard evaluation of our work. In the following, we have carefully read the questions raised by the reviewers, and reply to each question point by point.

Minor comments:

1. In the first paragraph, the authors mentioned the current challenges in the fabrication of superhydrophobic interfaces and proposed solutions. However, the link between the solutions and the challenges is not quite clear. To make their contribution clearer in terms of tackling the challenges, the authors can elaborate more on how their proposed solutions address the challenges.

Response:

We appreciate the reviewer for raising this good question. In the revised main text, we have added more on the links on how we propose the solutions to address the challenges. Please refer to “*In previous works, researchers have developed physical and chemical methods to prepare superhydrophobic surfaces, such as laser and template etching, electrochemical deposition and chemical etching, etc^{18,19}. Those methods can be classified into two categories: increasing the surface roughness and lowering the surface energy via chemical modification. However, the preparation of superhydrophobic cellulose film has faced the following problems. The first is that the physical etching can hardly be employed on the delicate cellulose film. The second is*

that cellulose fibers are hydrophilic in nature instead of being hydrophobic. Also it is very challenging to realize large-area, durable and low-cost superhydrophobic surfaces with good mechanical processability^{18,19}. To overcome these challenges, we resort to revolutionize the current paradigm. Firstly, robust superhydrophobic shielding must be introduced by decorating the cellulose fibers with nanoparticles to largely increase the surface roughness and meanwhile reduce the surface energy. Secondly, the fabricated superhydrophobic cellulose film should be compatible with high-precision laser-cutting for hollow-out acoustic metasurfaces of functionality customization.” on page 3, lines 46-60 in the revised manuscript.

2. Including a theoretical analysis of the transverse and longitudinal resolutions for the meta-lens may provide readers with a better understanding of the meta-lens's performance.

Response:

We thank the reviewer for this suggestion. In the revised supplementary materials, we add the Note 6 for analyses of the lateral and longitudinal resolutions in imaging.

Please refer to “*Resolution refers to the ability to distinguish two objects segregated in a distance. High resolution indicates that the imaging process can distinguish two objects with a small space. In ultrasound imaging, two types of resolutions are commonly used, viz., lateral and longitudinal resolutions. Lateral resolution, known as azimuth or radial resolution, describes the resolution perpendicular to the propagating axis of the beam. Longitudinal resolution, also known as range or axial resolution, describes the resolution along the propagating axis of the beam.*

The lateral resolution is mainly determined by the spot size of a focused ultrasound beam, as shown in Fig. S11(a). Here we set the spot size of the focused beam to be d_1 and the distance between two objects A and B also to be d_1 . When the focused beam is swept on the object A or B, there will be a strong echo signal received by the transducer. When the focused beam is swept on other areas, for example the space between objects A and B, the echo signal will be very weak. The images of objects A and B can thus be reconstructed through processing of echo signals. However, when the distance between

objects A and B is less than d_1 , the focused beam cannot be used to distinguish the two targets, where the echo signals will identify two objects as one.

Supplementary Fig. 11. Schematics of the lateral and longitudinal resolutions for ultrasound imaging based on meta-skis. a, The lateral resolution for distinguishing objects A and B . b, The longitudinal resolution for distinguishing interfaces A and B .

In the propagation direction, the smallest distance between two adjacent interfaces one can distinguish is termed the longitudinal resolution (ΔR_l). After the ultrasound is modulated by the meta-lens, it passes through the interfaces A and B in sequence and generates echo signals I (red) and II (blue), respectively, as shown in Fig. S11(b). When the echoes I and II do not overlap, we can distinguish interfaces A and B , for which the following formula must be satisfied

$$\tau \leq \frac{2d_2}{c}, \quad (\text{S21})$$

where τ is the pulse width, c is the sound velocity, and d_2 is the interfacial

separation. Thus the following formula can be obtained

$$d_{\min} = \Delta R_l = \frac{\tau c}{2}, \quad (\text{S22})$$

where d_{\min} is the smallest distance between two targets that can be distinguished, viz., the longitudinal resolution. When the distance between two interfaces is less than ΔR_l , the imaging system cannot distinguish the two interfaces and mistakenly identifies them as one single interface. The pulse width τ is the main factor influencing the longitudinal resolution.” in supplementary note 6.

3. The authors may consider to include a model to characterize the achieved meta-skin, for example, its acoustic parameters. Such a model can help readers understand the meta-skin's behavior and its potential applications.

Response: We sincerely appreciate the valuable suggestions provided by the reviewer. In the revised version of the manuscript, we have made the necessary additions based on these suggestions. (Supplementary Note 3. Pages 9-11)

Please refer to

“1. Model of superhydrophobic surfaces

Generally, the meta-skin surface is very rough due to superhydrophobic property. The surface roughness is conformed to the Gaussian distribution, which is expressed as²

$$\begin{aligned} R_f &= \frac{1}{A_{wf}} \iint \sqrt{1 + \left(\frac{\partial z}{\partial x}\right)^2 + \left(\frac{\partial z}{\partial y}\right)^2} dx dy \\ &= \sqrt{1 + 2 \frac{\sigma^2}{L^2} \times \frac{[\exp(-\frac{\beta}{L})]^2}{\pi}}, \end{aligned} \quad (\text{S1})$$

where R_f is the surface roughness, β is the correlation length, L is the characterization step length of the surface morphology, and σ is the standard deviation.

Supplementary Fig. 5. Surface roughness of BC meta-skin. We randomly chose two positions (I and II) on the meta-skin to characterize the surface roughness by using the scanning probe microscope.

From the measurement results of the scanning probe microscope (SPM-9700) in Fig. S5, we can extract the parameters in Eq. (S1). In this example, the measurement step length $L=0.2 \mu\text{m}$, the average surface roughness $R_f=1.9 \mu\text{m}$, and the standard deviation $\sigma=0.44 \mu\text{m}$. Using these parameters, the correlation length β is calculated to be almost zero, indicating that the meta-skin surface is rather rough, which is one of the necessary conditions for superhydrophobicity.

II. Simulation of total ultrasound reflection

Due to the superhydrophobic nature of the meta-skin, stable Cassie-Baxter states (viz., the micro-sized air bubbles) form on the rough nanosurface. We assume that the thickness of air layer in water is equal to the surface roughness of meta-skin. Therefore, in numerical simulations, we set the size of air microbubbles to be $1.9 \times 1.9 \mu\text{m}^2$ and the period of bubble array is $3.8 \mu\text{m}$ for a simplified model. The background is water. As shown in Fig. S6(a), total ultrasound reflection occurs at the air/water interface due to a large impedance mismatching in the frequency range from 0.3 to 5.4 MHz, with the

transmission less than -19.4 dB, in agreement with the experimental measurement. In Fig. S6(b), the calculated intensity field distributions show that the transmittance of ultrasound through the meta-skin decreases with increasing frequencies.

Supplementary Fig. 6. The reflection/transmission simulation. *a*, Transmission of ultrasound impinging on the BC meta-skin in the frequency range of 0.3~5.4 MHz. *b*, Intensity distributions of ultrasound at 0.2 MHz, 2 MHz, and 5.0 MHz. In the numerical simulation, the size of air microbubble is $1.9 \times 1.9 \mu\text{m}^2$ and the period of bubble array is $3.8 \mu\text{m}$. The ultrasound beams are incident from below and the transmission through the microbubble array is close to zero.” in supplementary note 3.

4. The authors should explain the abbreviation FWHM, which is not immediately clear to readers.

Response:

We appreciate the comment posed by the reviewer. In the revised version of the manuscript, we explicitly use the full term “Full Width at Half Maximum (FWHM)” to facilitate better understanding for the readers.

5. In the supplemental video, the image of the line under SIAT appears from left to right, while the image of the line under HUST appears from the middle to two sides. Is there any reason for these two different behaviors?

Response:

a

b

Fig. R5. Imaging objects “HUST” and “SIAT” used in the experiment.

We appreciate the reviewer’s careful observation and very meticulous question. In the Supplementary Video 6, the image of “SIAT” appears from left to right, and the image of “HUST” appears from the middle to two sides. In our experiment, the thickness of objects “HUST” and “SIAT” is only 100 μm , making the metallic objects flexible. During the assembling process, the objects “HUST” and “SIAT” are slightly distorted to a certain extent, as shown in Fig. R5. For example, the object “SIAT” has a slight inclination to the right side. The object “HUST” is bent with the central part curved toward the transducer. This slight distortion with an amplitude of several hundred micrometers can be distinctively rendered in the echo ultrasound signals, which also demonstrate the high longitudinal resolution of our approach. We add some discussions

on this part on page 24-25 in the supplementary materials.

6. The authors designed the holography and meta-lens for different frequencies, i.e., one is 500kHz and another is 5MHz. The authors may explain why they chose different frequencies and how this affects the performance of the holography and meta-lens.

Response:

Thank you for the good question raised by the reviewer. Here we chose 500 kHz and 5 MHz for ultrasound holography and 3D imaging to demonstrate that our meta-skin lens can modulate ultrasound in broadband. As for the imaging meta-skin lens, we chose the frequency of 5MHz for achieving a higher resolution. For the holographic meta-skin lens, we chose the frequency of 500 kHz for achieving a hologram field pattern in a large scale and further benefited the ultrasonic tweezing of large objects. The related discussion is appended on Page 17 in the revised main text.

7. In Fig 3 d-II, the simulation results show a single peak in the center without two side peaks, which is quite different from the theory and the experiment. The authors should explain this discrepancy.

Response:

We thank the reviewer for this good question. In the theoretical calculation, we focus on the case in free space without considering the boundary reflection. The experimental measurement was carried out in a water tank ($70 \times 120 \times 80 \text{ cm}^3$). Since the water tank is large enough, it can be regarded as the wave propagation in free space. In the 3D simulation, the overall model cannot be set very large due to the computation resource limitation. Therefore, the plane wave radiation boundary condition cannot provide a perfect absorption for obliquely incident waves, resulting in unwanted side peaks. In the revised main text, we add some discussions on this point on Page 11 in the revised main text.

Reviewer #3 (Remarks to the Author):

Main comments:

The paper presents a novel mechanism for transforming hydrophilic Bacterial Cellulose (BC) into a hydrophobic structure by carefully treating BC nanofibers with polydimethylsiloxane-coated SiO₂ nanoparticles. The authors claim that their process, which involves converting the BC hydrogel into a BC alcogel and subsequently treating it with SiO₂ nanoparticles, is a unique approach. This treatment results in a meta-skin with enhanced flexibility and machinability, enabling the creation of kirigami structures through laser cutting. Although I am not an expert in the specific fabrication process, I find the description to be both intriguing and clear. The proposed method has the potential to attract significant interest from researchers working on acoustic metamaterials and fabrication, who may further explore the design and application of these materials.

Response: Thank you for your careful review and constructive suggestions. In the following, we will try our best to answer reviewer's comments.

1. The presentation of results in Figure 2 is somewhat confusing. Figures 2a and 2g appear to be related, while Figures 2d, 2e, and 2f pertain to the same measurement, and Figures 2b and 2c form a third group of data. To improve clarity, the figures and their corresponding descriptions should be reorganized to present a more coherent set of results. For instance, Figures 2a and 2g could be grouped together, along with the relevant text that discusses these figures.

Response:

We thank the reviewer for this suggestion. We have grouped together the Figs. 2a and 2g along with the relevant text that discusses these figures (see Fig. 2. Page 8).

2. Additionally, to enhance the comprehensibility of Figure 3a, it would be beneficial to include axis labels, as seen in Figure 3c, and provide a clear indication of the origin (0,0,0).

Response:

Thank you for the suggestion. In the revised manuscript, we adopt the reviewer's advice to provide axis labels and a clear indication of the origin (0, 0, 0) in Fig. 3a. (Fig. 3. Page 10).

3. I find it unclear why the authors describe their structure as a "kirigami" structure. While the treated BC undergoes laser cutting to create a pattern of holes that functions as a transmissive meta-structure, I do not observe any folding in the design. In traditional kirigami structures, folding is typically intentional, incorporated during fabrication to achieve specific properties or functionalities. Thus, the term "kirigami" may not accurately represent the surface in this case.

Response:

Thank you for this question. In order to avoid potential misunderstandings, we replace "kirigami" with "paper-cutting" or "metasurface" in the revised version. We believe this change is more appropriate and we appreciate the reviewer for bringing up this valuable question. In our future work, we will focus on investigating 3D acoustic meta-devices based on Kirigami and Origami of the paper-like BC meta-skin.

4. I disagree with the authors' claim of good agreement between the theoretical, simulation, and experimental results. The presented data is not conclusive, as it relies on visual comparison, which is subject to various uncertainties, such as print resolution and choice of scale and colors. I suggest presenting the data as a 2D image (instead of the 3D perspective view shown in the paper) and employing similarity metrics like SSIM or MSE for comparison, rather than relying solely on visual inspection. While visual comparison is relevant and useful, additional metrics would strengthen the analysis.

Response:

Thank you for this question. In the revised main text, we present the data as a 2D image and employ the similarity metrics of MSE for comparison. We also elaborate the reason of unwanted side peaks of simulations in comparison with the theoretical and measured results. Please refer to "*In Fig. 3c, the theoretical, simulation and measured results are*

presented with 'H'-like distributions. Compared with the theoretical result, the mean square errors (MSEs) of the simulation and experimental results are 0.1294 and 0.1425, respectively. In Fig. 3d, we compare the intensity distributions on the planes I, II, III marked in Fig. 3c quantitatively, where the results show a close resemblance. It should be mentioned that in the theoretical calculation we focus on the case in free space without considering boundary reflection. The experimental measurement was carried out in a water tank ($70 \times 120 \times 80 \text{ cm}^3$). Since the water tank is large enough, the wave propagation can be regarded as in free space. In the 3D simulation, the model cannot be set very large due to the limitation of computation resource. Therefore, the plane wave radiation boundary condition cannot provide a perfect absorption for obliquely incident waves, resulting in unwanted side peaks.” on Page 11, lines 231-243.

5. The core argument and contribution of this paper seem to revolve around the creation of bio-active, superhydrophobic BC structures. Although using through-hole patterning to manipulate sound waves is not a novel concept, the paper demonstrates the potential of BC meta-skin for acoustic holography and meta-lenses, thereby providing proof of concept for its effectiveness in manipulating sound waves. I recommend that the authors either refrain from using the term "kirigami" or provide a clearer explanation for why their structure genuinely incorporates both folds and cuts.

Response:

Thank you for the valuable suggestion. In order to avoid potential misunderstandings, we replace “kirigami” with “paper-cutting” or “metasurface” in the revised version. We believe this change is more appropriate and we appreciate the reviewer for bringing up this valuable question. In our future work, we will focus on investigating 3D acoustic meta-devices based on Kirigami and Origami of the paper-like BC meta-skin.

6. Finally, I also feel that the paper is missing some key intermediate results/characterisations. Investigating the ability to manipulate the density of SiO₂ nanoparticles in the BC hydrogel can be beneficial for understanding and tuning various properties of the material. Some studies should be conducted to understand and

characterise this better. For example, what is the distribution and uniformity of the nanoparticles in the BC hydrogel?

It would be good to know more about the mechanical properties of the BC meta-skin (tensile strength, elastic modulus, and ductility) and how it changes with different SiO₂ density.

Response:

Thank you for the valuable suggestion. In the revised manuscript, the Fig. 1(d) shows that the nanoparticles are distributed uniformly on the BC fibers. Based on your advice, we also add mechanical properties (tensile strength, elastic modulus, and ductility) and contact angles with different SiO₂ densities.

Please refer to “

Supplementary Fig. 1. Superhydrophobicity and mechanical property of BC meta-skin. **a**, A water droplet impinging on the meta-skin, which was taken by a high-speed camera, indicating the existence of superhydrophobicity. **b**, SEM images of the BC meta-skin. **c**, The contact angle (CA) of a water droplet resting on the meta-skin surface. Within 200 days, the measurement of CA verified the stability of superhydrophobic

property of BC meta-skin with/without immersing in water. d, The CA of decorated BC meta-skins prepared with different SiO₂ concentrations. e, Young's modulus and tensile strength of decorated BC meta-skins with different SiO₂ concentrations. f, The strain-stress curves of pure BC membrane and decorated BC meta-skin.

In Figs. S1(d)-(f), we further study the relation between the material properties and the SiO₂ concentrations in preparation. In the experiments, the samples were prepared with dispersion solution under different SiO₂-nanoparticle concentrations of 0% (Pure BC membrane), 0.05%, 0.1%, 0.5%, 1% and 2.5% (BC meta-skin). The measured average CAs of the samples are shown in Fig. S1(d). The result shows that with the concentration of SiO₂ nanoparticles increasing from 0% to 2.5%, the CAs of samples change from 45.0° to 163.2°. It is notable that the CA becomes stable for the samples with SiO₂ concentration above 0.5%, which is at around 160° and clearly indicates the existence of superhydrophobicity. In this work, the 2.5% SiO₂-nanoparticle dispersing solution was utilized to fabricate the decorated BC meta-skin.

In Fig. S1(e), the measured results demonstrate that the mechanical properties of BC meta-skin are also enhanced by the decorative SiO₂ nanoparticles. To be specific, the Young's modulus and tensile strength of decorated BC meta-skins become larger as SiO₂ concentration increases, for which the average tensile strengths of pure BC membrane (0% SiO₂ concentration) and BC meta-skin (2.5% SiO₂ concentration) are 52.8 MPa and 59.9 MPa, respectively. In the tensile failure test, as shown in Fig. S1(f), the strain-stress curves of BC membrane and meta-skin show that the maximum stresses undertook by the same-size samples are 65.1 MPa and 77.7 MPa, respectively.

” on Pages 3-5 Supplementary Information.

7. The contribution of this paper is both novel and interesting. I feel it would make a nice contribution to Nature Comms. But I would also appreciate more rigorous testing and characterisation of the properties of the meta-skin before it can be included for publication.

Response:

Thank you for the positive evaluation. In the revised version of the manuscript, we have incorporated the reviewer's suggestions and conducted more rigorous testing and characterization of the material's properties, including tensile strength, elastic modulus, and ductility. We have also made corrections to any terms that might have caused misunderstandings. We sincerely appreciate the reviewer's constructive questions and suggestions.

Reviewer #1 (Remarks to the Author):

The authors have fully addressed my comments in the revised manuscript. With the added details on the properties and characterization of the metasurface, the manuscript demonstrates a meaningful approach for applications in biomedical fields. I am glad to see its publication.

Reviewer #2 (Remarks to the Author):

The authors have addressed my previous questions.

Reviewer #3 (Remarks to the Author):

The authors have addressed my earlier comments well and after reading the revised manuscript, i believe it is still a novel contribution that is worth including in Nature Comms.

Response Letter to Reviewers

We sincerely thank the three reviewers for their careful review of the manuscript and their constructive guidance, which were crucial to improving our manuscript. We thank the three reviewers for their patience and careful review.

Reply to the Reviewer #1

Main comments:

The authors have fully addressed my comments in the revised manuscript. With the added details on the properties and characterization of the metasurface, the manuscript demonstrates a meaningful approach for applications in biomedical fields. I am glad to see its publication.

Response:

We thank the reviewers for their careful review of the manuscript and recommendation of publication in Nature Comms.

Reply to the Reviewer #2

Main comments:

The authors have addressed my previous questions.

Response:

We thank the reviewers for their careful review of the manuscript and recommendation of publication in Nature Comms.

Reply to the Reviewer #3

Main comments:

The authors have addressed my earlier comments well and after reading the revised manuscript, i believe it is still a novel contribution that is worth including in Nature Comms.

Response:

We thank the reviewers for their careful review of the manuscript and recommendation of publication in Nature Comms.